# Fine-Grained Visual Understanding for Multimodal and Trustworthy AI

## Abstract

Fine-Grained (FG) and Ultra-Fine-Grained (UFG) Visual Understanding has recently become an important problem in AI research, because of its considerable ability to distinguish objects visually very similar but semantically different. This paper aims to offer a viewpoint-based overview and taxonomy on the state of the art of the FGVC tasks, covering existing FGVC datasets as well as approaches, and identify possible pros and cons for FGVC datasets in terms of scalability, cost basis of annotating, domain coverage and generalization. We also review how recent trends, with transformer-based vision architecture, advanced data augmentation (in particular generative augmentation), and the multimodal integration of vision, language/metadata are pushing towards the practical need for FGVC and Ultra-Fine-Grained Visual Categorization (UFGVC) necessary in building multimodal and trustworthy AI systems. In our study, we detect a number of standing challenges: lack of public ultra-fine-grained datasets, high annotation complexity, non-trivial long-tail/rare class learning setting and inadequate exploitation on multimodal and semantic context. Finally, we present a prospective research roadmap on multimodal visual understanding covering robustness under long-tailed distributions, explainability, data efficient learning and deployment to real-world applications. Motivated by previous advances, as well as the remaining challenges in this field, we hope that this survey can shed light on the design of more generalizable, reliable and semantically-grounded visual intelligence.

ACM Reference Format:
. . Fine-Grained Visual Understanding for Multimodal and Trustworthy AI. In *Proceedings of* . ACM, New York, NY, USA, 13 pages.

## 1 Introduction

This study will begin by elucidating the primary motive for interpreting the significance of Fine-Grained Visual Categorization (FGVC)/Ultra-Fine-Grained Visual Categorization (UFGVC) in relation to Multimodal and Trustworthy AI. Furthermore, it will provide definitions and delineate the scope, concluding with an introduction to the aims of this survey. Figure 1 summarizes our positioning of FG/UFG as evidence-centric primitives bridging multimodal alignment and trustworthy evaluation.

### 1.1 Motivation

This section examines one question: (I) Is FGVC / UFGVC considered an outdated question?

*1.1.1 Is FGVC / UFGVC considered an outdated question?* Recent developments suggest that FGVC and UFGVC are not out-of-favor, but are instead on an upswing within modern visual understanding, propelled by transformer-based architectures, improved learning

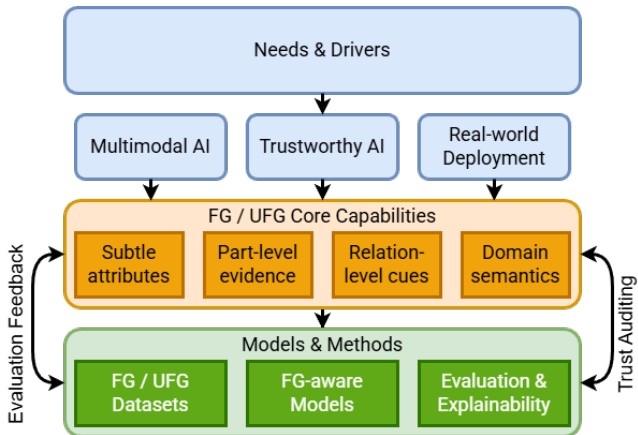

**Figure 1: Positioning of fine-grained (FG) and ultra-fine-grained (UFG) visual understanding within multimodal and trustworthy AI.**
**Real-world deployment demands multimodal alignment, trustworthiness, and reliability, motivating FG/UFG capabilities such as subtle attribute recognition, part-level evidence localization, and relation-level reasoning, which in turn shape datasets, model architectures, and evaluation.**

procedures, and broader real-world requirements. In FGVC, hierarchical attention and transformer-based methods continue to improve both performance and methodology [13, 25, 103]. UFGVC remains an open frontier and a significant challenge, especially when inter-class differences are extremely subtle or beyond human perceptual capability, with strong relevance to applications such as agriculture and biodiversity [109]. Transformer-style and contrastive learning techniques continue to rejuvenate representation learning in fine-grained and ultra-fine-grained settings [27, 106, 107], while attention-based and part-detection approaches still hold their ground across canonical benchmarks [9, 10]. The continued release of datasets and domain-specific deployments (e.g., plant cultivar or produce recognition) provide evidence of sustained community engagement and practical utility [88, 95, 109, 115]. Meanwhile, mask-based and self-supervised methods further enhance methodological diversity and sample efficiency [55, 56, 108].

### 1.2 Why FG / UFG Is Becoming More Important?

Fine-grained (FG) and ultra-fine-grained (UFG) perception is no longer just a "harder classification setting"; it is becoming a structural requirement for modern AI. This shift is driven by three converging trajectories—multimodal intelligence, trustworthy interpretability, and real-world deployment—where decisions increasingly hinge on subtle attributes, parts, and interactions. Figure 2

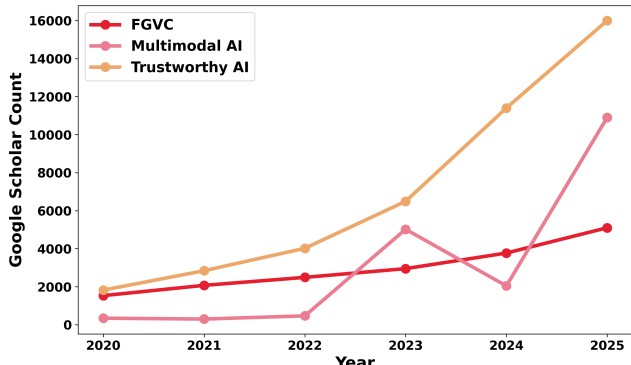

**Figure 2: From 2020 to 2025, FGVC shows steady growth, while Multimodal AI and Trustworthy AI surge rapidly in recent years, indicating a shift toward multimodal integration and reliable AI research.**

shows the publication-trend analysis under our exact query strings (Appendix A.4).

**Multimodal AI.** Contemporary multimodal systems depend on stable and interpretable fine-grained primitives to align representations across vision, language, and other modalities. Frameworks such as DIME and MultiSHAP suggest that disentangling *fine-grained* cross-modal interactions enables interpretable and robust reasoning, which is crucial for debugging and alignment in multimodal models [46, 90]. Moreover, FG analysis naturally supports semantic supervision in tasks where subtle attributes and relations determine performance—for example, emotion recognition and language–vision grounding [38]. As multimodal AI expands beyond vision–language into biosignals, speech, and robotics, fine-grained interpretability increasingly serves as the backbone for generalizable alignment across heterogeneous sensing and decision pipelines [44].

**Trustworthy AI.** As trustworthy and explainable AI becomes a central requirement, many fairness, bias, and reliability failures are found to originate at the fine-grained feature/attribute level rather than at coarse class labels. Accordingly, trust increasingly relies on interpretability methods that expose fine-grained causal and interactional structures, especially for black-box foundation models [75, 77]. In medical and social domains, human-centered trust is often achieved through explanations at the diagnostic-feature level—not merely through high-level confidence scores [21]. In addition, the rise of synthetic content makes fine-grained forensics (e.g., pixel-/token-level cues) essential for transparency and for countering deception, as exemplified by recent multimodal forensics models [39].

**Real-world Deployment.** FG/UFG capabilities have shifted from academic benchmarks to deployment necessities in safety-critical and high-precision settings. Deployment-centric multimodal AI highlights that fine-grained interpretability is required to ensure reliable operation across healthcare, environmental monitoring, and industrial applications [44]. For instance, multimodal ECG resources such as MEETI emphasize beat-level annotations to support trustworthy diagnosis and integration into clinical decision

support workflows [112]. Similarly, biodiversity and agriculture applications often require ultra-fine-grained recognition (e.g., subtype-level fungi identification), where such distinctions directly affect ecosystem management and agricultural health [58].

In summary, FG/UFG understanding is becoming indispensable because modern multimodal, trustworthy, and deployed AI systems increasingly depend on precise, interpretable, and ethically constrained handling of detailed semantic attributes. In high-stakes, human-centered settings, FG/UFG is where performance, reliability, and accountability ultimately converge.

## 1.3 Definitions & Scope

This survey adopts the position that *fine-grained* (FG) and *ultra-fine-grained* (UFG) problems should not be reduced to "a deeper taxonomy" or "more classes". While label granularity often correlates with difficulty, the defining challenge is the *kind of understanding* required to separate categories: correct recognition hinges on weak, localized, and compositional evidence that is easy to miss, easy to confuse with context, and often meaningful only under domain-specific semantics. Under this view, FG/UFG settings are best interpreted as instances of *fine-grained visual understanding*, where the model must ground decisions in subtle attributes, part-level structure, and relation-level cues rather than relying on coarse global appearance or dataset-specific shortcuts.

### 1.3.1 Clarifying FGVC, UFGVC, and Fine-Grained Understanding.

*Fine-Grained Visual Categorization (FGVC).* FGVC typically refers to distinguishing sub-categories within a shared super-class, such as species within birds or trims within a vehicle family. The difficulty arises from a characteristic imbalance: inter-class differences are small and often concentrated in limited regions, whereas intra-class variation induced by pose, illumination, viewpoint, occlusion, and background can be substantial. As a result, FGVC is not merely a matter of capacity or data scale; it is a matter of whether representations and training signals encourage attention to the truly discriminative evidence and suppress spurious correlations. In FG/UFG, such spurious cues often come from background or acquisition artifacts, so evaluations should include slice checks or simple perturbations that break non-causal context.

*Ultra-Fine-Grained Visual Categorization (UFGVC).* UFGVC narrows inter-class margins to the point where discriminative cues are faint, highly localized, and sensitive to imaging conditions. Consequently, reliable evaluation typically demands tighter protocols and high-fidelity annotations, since small labeling or acquisition ambiguities can dominate observed performance; the goal shifts from identifying the object to resolving its precise variant or condition under domain-specific semantics.

*Fine-grained understanding is more than category granularity.* FG/UFG refers to the level of visual understanding required rather than the number of labels. In practice, recognition depends on subtle visual attributes, localized part-level evidence, relational structure, and domain-specific semantics. Consequently, a task can be fine-grained even with few classes if such evidence is required, while a large label set does not necessarily imply fine-grained understanding when coarse cues suffice.

*1.3.2 Scope of this survey.* This survey covers datasets, methods, and evaluation protocols that operationalize FG/UFG recognition for multimodal trustworthy AI, focusing on how distinctions are defined, supervised, and assessed; how models attend to discriminative attributes/parts/relations; how language/metadata/other modalities ground subtle visual differences; and how reliability (shift robustness, long-tail behavior, attribute-/part-level faithfulness) is ensured. It is not leaderboard-centric; comparisons are used only to expose trade-offs (spurious cues, noise sensitivity, data efficiency, generalization, and evidential support).

*1.3.3 Working definition used throughout.* Throughout the remainder of the paper, we use the following operational definition to guide our perspective-driven analysis. *Fine-/ultra-fine-grained visual understanding* refers to a model's ability to make correct, robust, and explainable decisions when the decisive signal is subtle, localized, and compositional, and when the meaning of distinctions is grounded in domain semantics that may not be captured by generic object categories. This definition makes explicit why FG/UFG should be studied beyond label granularity, and it motivates the survey's emphasis on multimodal grounding and trustworthiness rather than benchmark ranking alone.

## 1.4 Survey Objectives and Contributions

Fine-Grained and Ultra-Fine-Grained Visual Categorization (FG/UFG) constitute a stringent testbed for modern visual recognition, where decisions rely on subtle, localized, and compositional evidence such as object parts, attributes, and fine-grained relations. Beyond accuracy benchmarks, FG/UFG expose fundamental limitations in robustness, calibration, and interpretability, particularly under distribution shifts and annotation ambiguity.

This survey has three objectives: (i) to organize recent FG/UFG advances from the perspectives of multimodal alignment and trustworthy AI, (ii) to identify structural challenges in evaluation and deployment beyond data scarcity, and (iii) to outline future research directions.

Accordingly, we present three complementary perspectives: (i) FG/UFG as semantic primitives that support fine-grained grounding and multimodal alignment beyond global image–text matching; (ii) FG/UFG as reliability stress tests that reveal robustness and calibration failures under uncertainty and distributional shift; and (iii) FG/UFG as tools for auditing and transparency, enabling attribute- and part-level explanations for bias analysis and accountable decision-making.

## 2 Background

### 2.1 Historical Evolution

From the earliest handcrafted visual systems to modern Vision-Language Models (VLMs), the development of FG and UFG recognition reflects the changing way in which models extract and align evidence. Early methods explicitly defined primitives and relied on human-engineered descriptors, while deep learning gradually internalized these processes, embedding them into more complex and integrated architectures. Yet, despite this shift, the core demand for fine-grained discrimination — recognizing subtle, localized cues

— has never disappeared, only been increasingly hidden within larger representational and alignment frameworks.

**Handcrafted.** Initial FGVC methods were dominated by hand-engineered features like SIFT, HOG, and BoW that explicitly defined visual primitives. These models relied on manually selected keypoints and descriptors to represent fine details in texture and shape [29]. While interpretable, such systems lacked adaptability and performed poorly on large-scale, complex datasets.

**CNN.** The emergence of Convolutional Neural Networks (CNNs) revolutionized FG recognition by learning localized discriminative features automatically. CNN-based architectures such as ResNet and attention-augmented models could extract hierarchical texture cues and region-specific signals, but they often over-relied on shortcuts and context biases rather than truly fine-grained evidence [26, 68].

**Transformer.** Transformers reframed FG recognition by making evidence selection itself the central task. Their self-attention mechanism enables global reasoning about which visual tokens matter, bridging local and contextual cues [27, 87]. This paradigm integrates both local and global dependencies but often turns discriminative "parts" into implicit attention maps rather than explicit primitives.

**Multimodal.** In the Multimodal era, tasks like FG and UFG recognition are no longer treated as standalone problems but as emergent capabilities of massive image–text alignment systems such as CLIP. Models like FGM-CLIP show that multimodal pretraining can capture fine-grained cues through implicit cross-modal correlations, though this also makes errors appear more "human-like" [40, 97].

Across this evolution, FG/UFG recognition has transitioned from explicit manual design to implicit representational learning. The need for subtle, part-based reasoning persists, but it has been progressively absorbed into broader architectures where "what counts as evidence" is learned, weighted, and sometimes obscured. This marks not the disappearance of fine-grained vision, but its deep embedding into multimodal intelligence.

### 2.2 FG / UFG as Perceptual Bottlenecks

Recent work on fine-grained and multimodal perception suggests an intermediate *representational bottleneck* in which compression attenuates subtle yet semantically decisive cues, especially when weak visual evidence must be aligned with high-level concepts [43, 96, 104]. In this view, FGVC/UFGVC function less as "finer classification" and more as diagnostic probes: they test whether internal representations retain minimal discriminative evidence for semantic grounding under overlapping cues and annotation ambiguity. For MLLMs, such bottlenecks often manifest as encoder–reasoner misalignment, degrading reasoning under subtle distribution shifts [24, 54]. Across FG/UFG and multimodal studies, models near this bottleneck repeatedly exhibit evidence–concept misalignment, unsupported attribute hallucination, and shortcut reliance on spurious correlations rather than discriminative cues [2, 67, 80].

Importantly, these failure modes have been independently reported across different architectures and tasks, including CLIP-like models, concept bottleneck models, and recent MLLMs, suggesting that they reflect structural limitations rather than isolated implementation issues.

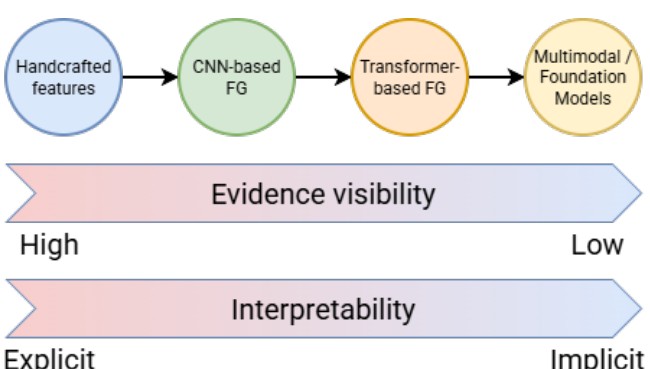

Figure 3: Historical evolution of FG and UFG visual recognition. The progression from handcrafted features to CNNs, transformers, and multimodal foundation models reflects a shift from explicit, human-defined evidence toward implicit, learned representations. While modern models achieve strong performance, the visibility and auditability of fine-grained evidence often decrease, motivating renewed emphasis on interpretable FG/UFG structures.

Ultimately, the FG/UFG perceptual bottleneck should be reframed as a design insight rather than a limitation: it highlights the need for models that can disentangle minimal sufficient representations, align semantics across modalities, and verify evidence reliability. Thus, FG/UFG benchmarks function not merely as finer recognition tasks, but as semantic atoms—essential testing grounds and alignment interfaces for trustworthy multimodal cognition and system design [11, 69].

This dataset-level trend supports the same interpretation. As shown in Figure 4, recent benchmarks increasingly target ultra-fine distinctions, implying that FG/UFG tasks are often used as diagnostic probes for representational bottlenecks rather than simply harder classification settings.

From a survey perspective, FG/UFG tasks therefore act as empirical probes of how perceptual evidence is selected, compressed, and aligned with semantics in modern vision and multimodal systems. This view consolidates evidence across datasets and models, clarifying why FG/UFG benchmarks are repeatedly adopted as stress tests for robustness, grounding, and trustworthiness.

## 3 Datasets and Methodologies

*Survey protocol.* We conducted a structured search over major scholarly indices (details in Appendix A), using predefined keyword sets and year filters, followed by title/abstract screening and full-text inclusion criteria to curate the reviewed papers. Exact search queries (used for Fig. 2) and the final paper list are provided in Appendix A for reproducibility. The dataset inventory in Table 1 (and Fig. 4 derived from it) is representative rather than exhaustive, hence temporal trends should be interpreted as indicative. We summarize commonly used evaluation metrics/protocols in Appendix B. Limitations include potential coverage bias from search

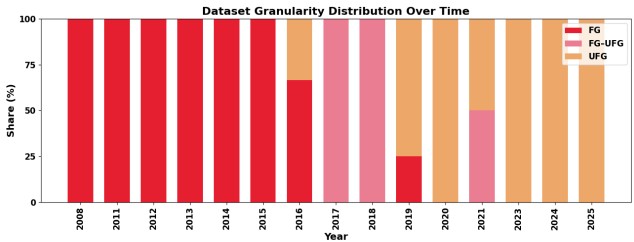

Figure 4: Dataset granularity has shifted over time: fine-grained (FG) visual recognition moved from early reliance on classical FG datasets toward increasingly ultra-fine-grained (UFG) settings. This reflects rising needs for subtle, expert-defined distinctions and a dataset-design evolution toward real-world semantics and deployment constraints, not just larger label sets. Based on Table 1; the dataset list is representative, so apparent trend changes should be interpreted cautiously.

sources/keywords and subjective boundaries between FGVC and UFGVC.

From the perspective of FG/UFG as a semantic bottleneck, datasets are not merely collections of labels, but mechanisms that decide which fine-grained evidence can be learned, audited, and aligned across modalities. The historical progression of dataset granularity shown in Figure 4 provides important context for understanding why contemporary FG/UFG benchmarks differ fundamentally from earlier fine-grained datasets.

### 3.1 FG Datasets

Classical FG datasets such as ImageNet[63] or CUB-200[85] have been instrumental in advancing visual recognition, but their apparent accuracy saturation does not mean the problem is solved. Studies show that high accuracy often reflects dataset-specific biases or limited variability, which artificially inflates performance metrics without true generalization [94]. For example, vehicle classification models achieve over 95% accuracy but fail under more diverse or realistic testing conditions due to inherent dataset homogeneity [73]. This accuracy saturation thus reflects overfitting to narrow domains rather than progress toward robust, trustworthy AI systems.

These limitations of classical FG datasets motivate a push toward UFG settings, where distinctions are more semantically precise but also far more demanding in terms of annotation fidelity, expert involvement, and ambiguity management.

### 3.2 UFG Datasets

UFG datasets push toward higher resolution and deeper semantic distinctions, yet they come with high annotation costs, subjectivity, and dependence on domain experts. For instance, the TomatoMAP dataset for plant phenotyping required extensive manual labeling by experts to ensure consistency across growth stages [113, 114]. Similarly, fine-grained legal datasets rely on human experts to label fact-article correspondences, revealing that fine distinctions

**Table 1: Representative FG and UFG Visual Recognition Datasets, sorted chronologically by release year. Beyond cataloging datasets by domain and granularity, the table highlights a structural transition toward ultra-fine-grained, expert-driven, and long-tailed benchmarks, underscoring emerging challenges in annotation cost, generalization, and trust-related evaluation discussed in later sections.**

| Dataset | Year | Venue / Source | Domain | Granularity | Task Type |
|---|---|---|---|---|---|
| Oxford Flowers 102 | 2008 | ICVGIP 2008[53] | Flowers | FG (category-level, 102 classes) | Classification |
| Caltech-UCSD Birds-200-2011 (CUB-200-2011) | 2011 | Dataset release[85] | Birds | FG (species-level, 200 classes) | Classification / Part localization |
| Stanford Dogs | 2011 | FGVC @ CVPR 2011[34] | Dogs | FG (breed-level, 120 classes) | Classification |
| Leafsnap | 2012 | ECCV 2012[37] | Plants / Trees | FG (species-level) | Classification |
| FGVC-Aircraft | 2013 | arXiv 2013[47] | Aircraft | FG (variant / model-level, hierarchical) | Classification |
| Stanford Cars (Cars196) | 2013 | FGVC @ CVPR 2013[36] | Cars | FG (make / model / year) | Classification |
| Birdsnap | 2014 | CVPR 2014[4] | Birds | FG (species-level, 500 classes) | Classification |
| Food-101 | 2014 | ECCV 2014[7] | Food | FG (dish-level, 101 classes) | Classification |
| NABirds | 2015 | CVPR 2015[81] | Birds | FG (species-level with subspecies / gender) | Classification |
| CompCars | 2015 | CVPR 2015[102] | Cars | FG (model-level; web + surveillance) | Classification / Verification |
| LifeCLEF / PlantCLEF 2015 | 2015 | LifeCLEF Challenge[20] | Plant Identification | FG (species-level, ~1000 classes) | Classification / Retrieval |
| PlantVillage | 2015 | arXiv 2015[31] | Agriculture (Plant disease) | FG (species × disease) | Classification |
| DeepFashion | 2016 | CVPR 2016[45] | Fashion | FG (categories / attributes) | Classification / Retrieval |
| Stanford Online Products (SOP) | 2016 | CVPR 2016[70] | E-commerce Products | UFG (instance-level) | Retrieval / Metric Learning |
| Urban Trees (Pasadena Urban Trees) | 2016 | CVPR 2016[91] | Urban forestry / Remote sensing | FG (species-level) | Detection + Classification |
| iNaturalist 2017 (iNat2017) | 2017 | CVPR 2018[83] | Biodiversity | FG–UFG continuum (taxonomy-aware) | Classification / Detection |
| HAM10000 | 2018 | arXiv 2018 / ISIC Challenge[12, 78] | Medical (Dermatology) | FG–UFG continuum (lesion-type level) | Classification / Segmentation |
| DeepFashion2 | 2019 | CVPR 2019[16] | Fashion | UFG (clothing identity with dense annotations) | Detection / Pose / Segmentation / Re-ID |
| Herbarium-2019 | 2019 | CVPR 2019[74] | Herbarium Sheets / Botany | FG (species-level) | Classification |
| SKU-110K | 2019 | CVPR 2019[22] | Retail Shelves | UFG (dense instance-level detection) | Detection |
| IP102 | 2019 | CVPR 2019[95] | Insect Pests / Agriculture | FG (species-level) | Classification |
| Plant-Pathology-2020 | 2020 | arXiv 2020 / CVPR 2020 FGVC Workshop[76] | Apple Leaves / Plant Disease | UFG (fine-grained disease-level) | Classification |
| Products-10K | 2020 | arXiv 2020 / ICPR Challenge[3] | E-commerce Products | UFG (SKU-level) | Classification |
| Google Landmarks Dataset v2 (GLDv2) | 2020 | CVPR 2020[92] | Landmarks | UFG (instance-level) | Recognition / Retrieval |
| Danish-Fungi-2020 (DF20) | 2020 | WACV 2022[57] | Wild Fungi / Biodiversity Monitoring | UFG (species-level, long-tailed) | Classification |
| iNaturalist 2021 (iNat2021) | 2021 | CVPR 2021[82] | Biodiversity | FG–UFG continuum (taxonomy-aware) | Classification / Detection |
| UFGVC | 2021 | ICCV 2021[109] | Leaf | UFG (gene, order) | Classification |
| BIOSCAN | 2023 | NeurIPS 2023[18] | Insects | UFG (BIN, order, family) | Classification |
| AMI | 2024 | ECCV 2024[33] | Wild Insects / Camera Trap Monitoring | UFG (species-level, long-tailed, OOD) | Classification |
| AquaMonitor | 2025 | arXiv 2025[32] | Aquatic Invertebrates / Biodiversity Monitoring | UFG (species-level, ultra-fine-grained) | Classification |
| TomatoMAP | 2025 | arXiv 2025[113, 114] | Agriculture / Plant Phenotyping | UFG (phenotyping) | Classification / Detection / Segmentation |

improve interpretability but increase annotation subjectivity and labor intensity [15]. These challenges highlight the trade-off between dataset granularity and scalability, especially in domains requiring expert validation.

## 3.3 Hidden Gap: Dataset Design vs Trustworthy AI

While increasing dataset granularity improves fine-grained recognition performance, it does not automatically translate into trustworthy behavior. A deeper gap remains between how datasets are constructed and how trust-related properties—such as bias, explanation faithfulness, and uncertainty—are evaluated. Despite progress, a hidden gap remains between current dataset design and the goals of trustworthy AI. Most datasets still lack mechanisms for systematic bias analysis or explanation evaluation [1]; they fail to capture how demographic, environmental, or stylistic biases propagate through models [19]. Recent works emphasize that dataset documentation and hybrid bias-tracing frameworks are essential to bridge this gap, enhancing explainability and fairness in AI pipelines [64]. Without such integration, even highly accurate systems risk being untrustworthy due to unexamined biases and opaque evaluation protocols.

**In summary**, although FG and UFG datasets improve benchmark performance, they fall short of supporting trustworthy AI, as limited data coverage often leads to benchmark-specific gains rather than real-world generalization.

## 4 FG / UFG for Multimodal AI

While datasets determine which fine-grained cues are learnable, effective reasoning ultimately depends on aligning these visual distinctions with other modalities, especially language.

## 4.1 Fine-grained Visual–Language Alignment

Recent works emphasize that fine-grained alignment between visual and textual modalities is essential for improving reasoning accuracy in multimodal systems. For instance, VideoGLaMM achieves pixel-level grounding between video frames and text, effectively connecting temporal and spatial elements for precise part–phrase correspondence [50]. Similarly, AlignCAT introduces category- and attribute-based matching to refine weakly supervised visual grounding, enabling more accurate attribute grounding and part–phrase alignment [89]. Failure analyses, such as those from LEGO Co-builder, reveal that even advanced models like GPT-4o fail at detailed spatial reasoning tasks, exposing persistent limitations in fine-grained visual understanding [30]. Likewise, ViGoR demonstrates that large vision-language models (VLMs) still hallucinate nonexistent visual elements, but fine-grained reward modeling can mitigate such failures by reinforcing accurate grounding [100].

## 4.2 FG Knowledge as Multimodal Anchors

FG knowledge acts as symbolic hooks and reasoning anchors that connect perceptual inputs with conceptual structures. $M^2$ConceptBase exemplifies this by introducing a concept-centric multimodal knowledge base that links visual and linguistic representations through

context-aware symbol grounding, improving reasoning and retrieval performance [111]. Similarly, Dr-LLaVA leverages symbolic clinical grounding to constrain VLM reasoning with structured medical knowledge, ensuring interpretability and domain reliability [72]. Further, VaLiK demonstrates how aligning visual features to language for multimodal knowledge graph construction enhances reasoning depth in large models, making symbolic FG knowledge a key anchor for multimodal understanding [42]. Collectively, these advances show that FG knowledge structures can serve as cognitive scaffolds—stabilizing multimodal reasoning and reducing hallucination by tying perception to symbolic semantics.

However, improved multimodal alignment alone does not guarantee reliability or accountability, motivating a dedicated discussion on trustworthiness.

## 5 FG / UFG for Trustworthy AI

Trust failures in multimodal AI often occur when FG/UFG evidence is absent, confounded, or unverifiable. Although Table 1 categorizes datasets by granularity and domain, it does not capture trust-related properties. To address this, Table 2 re-examines FG/UFG datasets using *operational, reproducible* criteria: **Bias/Fairness** (subgroup annotations + auditing protocol), **Long-tail** (imbalance + rare-slice/OOD evaluation), **Annotation Quality** (expert vs. mixed pipelines), **Ambiguity** (uncertainty/disagreement representation), and **Explainability Signal** (strongest supervision such as parts/ hierarchy/lesions). We map each attribute to *High/Medium/Low* via a simple rubric: High = explicit annotations & protocol, Medium = partial/proxy support, Low = none. We further synthesize methods under a unified taxonomy, highlighting shared design choices, trade-offs, and failure modes across paradigms.

### 5.1 Bias and Fairness

Bias often hides in fine-grained attributes that coarse-grained labels fail to expose. Recent research demonstrates that fine-grained semantic computation allows AI systems to detect subtle biases across demographic groups by analyzing claim-level meanings rather than token-level features [99]. A practical limitation is that fairness cannot be directly audited when subgroup annotations are missing, so many FG/UFG studies rely on proxy slices as a minimum diagnostic rather than a complete fairness evaluation. Fine-grained fairness auditing frameworks such as Predictive Representativity further reveal outcome-level inequities, emphasizing the need to evaluate model generalization across underrepresented populations rather than merely balancing datasets [49]. Additionally, methods like Fairness Regularizers improve performance across minority subpopulations in long-tailed and noisy data scenarios, promoting equitable learning without sacrificing accuracy [92].

### 5.2 Transparency and Explainability

Fine-grained representations naturally support part-based explanations and attribute-level reasoning, enabling transparent and interpretable AI behavior. Representation Engineering (RepE) enhances AI transparency by analyzing high-level, population-based representations to better interpret complex model cognition [116]. Similarly, prototypical and self-explainable classifiers, such as Pantypes, capture diverse latent features to provide interpretable, part-level

justifications for model predictions [35]. Involving domain experts in the representation debiasing process can further enhance interpretability and fairness without reducing model accuracy [6].

### 5.3 Robustness and Long-tail Reliability

Long-tailed settings can further amplify spurious correlations, where head-class context becomes a shortcut that fails under rare-slice or shifted conditions. Long-tail scenarios represent ultra-fine distinctions where foundation models often exhibit their greatest fragility. Studies indicate that robust long-tail learning frameworks, such as ViRN and Distributional Robustness Loss, improve model generalization and reduce overfitting to head classes by enhancing representation quality for rare or fine-grained instances [14, 65]. Moreover, fine-grained evaluation frameworks like SALTED uncover rare but critical long-tail errors in generative and translation models, providing diagnostic visibility into subtle reliability issues [62]. These findings reinforce that ultra-fine granularity in representation and monitoring is central to building AI systems that remain fair, interpretable, and reliable under distributional stress.

## 6 Open Challenges and Research Gaps

Despite major advances in multimodal large language models (MLLMs), several open challenges remain in achieving FG and UFG multimodal understanding.

### 6.1 Lack of FG-aware multimodal benchmarks

Current benchmarks often lack the resolution and annotation richness necessary for FG/UFG analysis. New datasets such as FG-BMK and FineBadminton emphasize multi-level semantic hierarchies and spatio-temporal reasoning but still face issues in defining precise "evidence units" like parts, attributes, or relations [28, 105].

Emerging multimodal benchmarks increasingly recognize the importance of fine-grained evidence but still expose critical integration gaps. Recent work such as VER-Bench and FAVOR-Bench shows that models struggle to reason over subtle visual or temporal cues [59, 79]. Multimodal datasets like MACSA and Fakeddit demonstrate the value of linking textual, visual, and contextual elements for fine-grained reasoning and annotation richness [51, 101]. Moreover, biosignal-based studies highlight new directions for modeling embodied ambiguity and subjectivity in evidence interpretation [52]. Together, these advances underscore the need for FG-aware benchmarks that model multi-modal "evidence units" dynamically across perception, context, and interpretation.

### 6.2 Evaluation beyond accuracy

Recent studies show that accuracy-based metrics are insufficient for evaluating fine-grained reasoning and structural understanding in multimodal models. Process-oriented benchmarks such as MM-MATH assess intermediate reasoning steps and solution processes to reveal procedural and diagram-level errors [71]. Human-Aligned Bench incorporates human performance baselines to measure reasoning gaps between models and people [60]. This remains unresolved because we still lack standardized, evidence-centric protocols that separate genuine fine-grained reasoning from benchmark shortcuts, and in UFG settings even slight annotation or acquisition noise can dominate measured gains.

**Table 2: Trustworthiness Dimensions in FG / UFG Datasets**

| Dataset | Bias / Fairness | Long-tail | Explainability Signal | Annotation Quality | Ambiguity |
|---|---|---|---|---|---|
| CUB-200-2011 | Low | Low | Parts | Expert | Low |
| iNat2021 | Medium | High | Taxonomy | Mixed | Medium |
| BIOSCAN | Medium | Ultra High | Hierarchy | Expert | High |
| HAM10000 | High | Low | Lesion-level | Expert | Medium |
| AMI | Medium | Ultra High | OOD / Long-tail | Expert | High |

## 6.3 Integration with foundation models

While foundation models dominate multimodal AI, their FG/UFG integration is limited. Approaches like *HEMM* [41] and *SciVer* [86] show the need for "FG-aware adapters" or modular probes to enhance foundation models rather than retraining them entirely. This challenge persists because injecting FG-aware components into foundation backbones can disturb their learned global alignment, and in UFG a tiny shift in token-level attention can flip the predicted subtype. Integrating foundation models is non-trivial because FG/UFG needs evidence-level control (parts/attributes/relations), while general-purpose models provide limited hooks to enforce such fine-grained supervision.

## 6.4 Human-in-the-loop and expert knowledge integration

Few studies integrate expert reasoning into training or evaluation. Datasets like *EVADE* [98] and FineBadminton demonstrate the benefits of human refinement pipelines, yet comprehensive frameworks for continuous expert feedback loops in multimodal reasoning remain scarce. This gap is also economic: expert labels are costly, making label-efficient expert-in-the-loop strategies a key practical direction. It is still open because expert feedback is scarce, heterogeneous, and hard to translate into consistent training signals, and in UFG genuine inter-expert disagreement is often intrinsic rather than a fixable labeling error.

## 6.5 FG/UFG × Temporal or Process Understanding

Fine-grained temporal reasoning remains an underexplored domain. *TemporalBench* [8], *EOC-Bench* [110], and *VideoMathQA* [61] highlight persistent gaps in modeling ultra-fine temporal evidence — understanding process-level causality, motion sequences, and temporally entangled multimodal events. This gap remains because temporal evidence is sparse and entangled with context, making causality difficult to verify, and in UFG a few frames or a slight phase shift can decide the class.

## 7 Future Research Directions

This section outlines seven future research directions, synthesizing the preceding discussions to highlight promising paths for advancing fine-grained and ultra-fine-grained visual understanding in multimodal and trustworthy AI. Future benchmarks should define FG evidence units across modalities (image, text, metadata, biosignals) and incorporate controlled ambiguity and expert disagreement rather than enforcing single labels [48].

## 7.1 FG-aware Multimodal Representation Learning

Building upon the challenges identified in Section 6.1, we focus on FG evidence-preserving multimodal representations. Recent multimodal representation learning, especially vision–language pretraining, achieves strong performance on coarse-grained tasks such as retrieval and captioning. However, multiple studies indicate that this success does not reliably transfer to FG and UFG understanding, where decisions depend on subtle, localized, and compositional evidence. In many cases, FG/UFG capability emerges only incidentally, as standard pretraining objectives do not explicitly require such evidence to be preserved.

A key recurring limitation lies in the *unit of alignment*. Most methods align global image representations with global sentence embeddings, which encourages semantic averaging and allows discriminative part-, attribute-, or relation-level cues to be diluted. As a result, models may rely on convenient correlated signals, such as background context or acquisition artifacts, while remaining strong on conventional benchmarks. In addition, treating language as the default alignment interface constrains extensibility, since other modalities (e.g., sensor signals or structured metadata) do not naturally reduce to text. This is especially critical for temporal reasoning, where global-to-global alignment can average out short-lived discriminative moments unless temporally localized evidence units are modeled.

Recent work has begun to explore finer-grained alignment strategies, including part- or attribute-level matching, lightweight intermediate structures, and objectives that emphasize minimal visual differences. Nevertheless, these efforts remain fragmented, and a unified framework for preserving fine-grained evidence across modalities is still lacking.

## 7.2 FG for Bias Auditing and Model Diagnosis

Building upon the challenges identified in Section 6.2, we develop FG-aware evaluation and diagnostic protocols beyond accuracy. Recent studies show that many biases and failure modes in multimodal models manifest at the FG level. A model can appear fair under coarse evaluation yet make systematic errors on semantically meaningful slices defined by attributes, subtypes, or acquisition factors (e.g., pose, illumination, sensor). Such hidden heterogeneity means aggregate metrics may mask consistent harms.

Current bias audits often rely on coarse labels and broad groupings, which miss attribute-conditional bias and FG shortcuts. Diagnostic tools also provide limited FG resolution: saliency maps are frequently non-causal, and post-hoc explanations are hard to compare across samples. As a result, recent work calls for FG-aware auditing (e.g., attribute/part-based slicing, worst-slice reporting,

counterfactual perturbations, and evaluation against FG annotations), but standardized FG-level auditing frameworks remain underdeveloped.

## 7.3 Data-efficient & Self-supervised UFG

Building upon the challenges identified in Section 6.4, we revisit data-efficient and self-supervised learning for UFG settings under a realistic assumption: expert supervision is scarce, expensive, and imperfect. Our framework therefore does *not* treat expert labels as oracle signals, but as limited and potentially noisy supervision that must be modeled and allocated carefully.

In practice, UFG annotations often exhibit *inter-expert disagreement* due to subtle cues and *intrinsically* fuzzy category boundaries; this ambiguity is frequently a property of the task rather than a labeling defect. Consequently, resource-efficient pipelines emphasize (i) *active learning* and triage to send experts only the "hard cases", and (ii) *ambiguity-aware labeling* (e.g., multi-label or soft labels) to preserve uncertainty instead of forcing a single ground truth. Meanwhile, standard SSL objectives can be misaligned with UFG needs because broad invariances may suppress fine discriminative evidence, motivating UFG-compatible SSL variants that better respect near-neighbor distinctions.

## 7.4 FG as Interfaces between Perception and Reasoning

Building upon the challenges identified in Sections 6.3 and 6.5, we enable structured FG reasoning and temporal understanding with foundation models. A recurring limitation of current multimodal systems lies in the weak coupling between perception and reasoning. Visual encoders produce dense and expressive representations, while reasoning modules operate over abstract concepts and language. When the connection between these stages is implicit, models may generate coherent reasoning while relying on incorrect or unverified perceptual evidence, leading to brittle or misleading conclusions.

Recent work suggests that FG and UFG structures can serve as an explicit interface between perception and reasoning. Parts, attributes, relations, and simple processes provide inspectable and compositional primitives that translate raw sensory inputs into reasoning-ready representations. By making evidence selection explicit, such interfaces enable reasoning modules to operate on grounded information rather than opaque embeddings.

Despite growing interest in structured intermediates and neuro-symbolic hybrids, existing approaches remain fragmented and task-specific. A systematic treatment of FG primitives as a general-purpose interface—supporting evidence verification, ambiguity handling, and FG-aware evaluation—remains largely unexplored, highlighting a key direction for future multimodal research.

## 7.5 Summary: A Research Roadmap

Together, these four threads outline a roadmap for fine-grained vision–language research. FG-aware representation clarifies *what is preserved* by retaining semantically meaningful details. FG for auditing clarifies *what can be trusted* by making grounding and evidence inspectable. Data-efficient UFG clarifies *what can be learned* under limited supervision via transfer and weak/scalable signals.

FG as an interface clarifies *what can be reasoned about* by enabling compositional, verifiable primitives. Collectively, this agenda moves the field from plausible multimodal outputs toward systems that preserve the right semantics, justify claims, learn with fewer labels, and reason over grounded structure. Future multimodal FG/UFG research highlights: (1) semantically transparent benchmarks for ambiguity tolerance; (2) evaluation-aware architectures; (3) structural alignment with base models; (4) expert-in-the-loop learning and advisement, and (5) temporal reasoning to link perceptual and conceptual levels.

## 8 Conclusion

FG and UFG visual understanding should be treated as core foundations for general and trustworthy multimodal AI rather than niche recognition tasks. By forcing models to rely on subtle, localized, and compositional evidence, FG/UFG exposes whether multimodal systems are genuinely grounded or merely generating plausible outputs, while providing precise semantic building blocks for robust cross-modal alignment and faithful multimodal reasoning. At the same time, FG/UFG serves as a practical stress test for trustworthiness: when discriminative cues are weak, confounded, or scarce, failures in robustness, calibration, and uncertainty estimation become more visible, revealing shortcut learning, hallucination, and attribute-driven bias. Meaningful progress therefore requires a shift from prediction-centric pipelines to evidence-centric paradigms that preserve decisive traits, support transparent verification, and enable learning under expensive or ambiguous supervision. Crucially, closing the dataset sufficiency gap—in both coverage and realism—is necessary to move trustworthy FG/UFG beyond a small set of canonical benchmarks and toward multimodal models that generalize under distribution shift, communicate uncertainty responsibly, and remain auditable in deployment. Our coverage is limited by the availability of public datasets and prior literature, and by unavoidable subjectivity in delineating FG/UFG across domains; Appendix C details these scope choices.

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

## A Systematic Search Protocol

In order to limit bias and provide reproducible approach to the field, a systematic search strategy was implemented according to the predefined research questions. The review is organized over the following three Research Questions (RQs) to address conceptual demarcations, methodological compromises and evaluation validity of fine-grained visual understanding.

### A.1 Research Questions

- **RQ1: Operationalization of Definitions.** How are the definitions and operational boundaries of Fine-Grained (FGVC) versus Ultra-Fine-Grained Visual Categorization (UFGVC) established in the literature? specifically, how is the distinction operationalized beyond mere label granularity?

- **RQ2: Methodological Paradigms and Trade-offs.** Under what experimental settings are distinct methodological paradigms (e.g., CNNs, Transformers, VLMs, Generative, Metric Learning, and Part-based approaches) effective, and what are their associated deployment costs and failure modes?

- **RQ3: Impact of Data and Evaluation.** How do dataset characteristics (e.g., annotation quality, long-tail distributions) and evaluation protocols influence the validity, reproducibility, and generalizability of conclusions in fine-grained research?

### A.2 Sources

To ensure a comprehensive coverage of the literature, we conducted a systematic search across multiple primary scholarly databases and indexing services. The primary sources included **Google Scholar**, **arXiv**, **IEEE Xplore**, **ACM Digital Library**, **Springer Link**, **Elsevier (ScienceDirect)**, and **OpenReview**. These platforms were selected to encompass both established peer-reviewed venues and high-impact preprints, reflecting the rapid pace of advancement in the field. The final search across all databases was completed on **January 10, 2026**.

## A.3 Time Window + Filters

**Time Window (2013–2025):** We restricted our primary survey scope to the period between 2013 and 2025. This window was chosen to capture the complete evolution of modern Deep Learning approaches in Fine-Grained Visual Categorization (FGVC), starting from early CNN-based part-localization methods (circa 2013–2014) through the Transformer era, up to the emergence of current Multimodal Large Language Models (MLLMs).

**Inclusion and Exclusion Criteria:**

- **Publication Type:** We prioritized peer-reviewed articles from top-tier computer vision and machine learning conferences (e.g., CVPR, ICCV, ECCV, NeurIPS, ICML, AAAI) and journals (e.g., TPAMI, IJCV). Given the high velocity of research in Multimodal AI, we also included impactful preprints from arXiv and OpenReview that have garnered significant community attention or citations.
- **Language:** The search was restricted to manuscripts written in English.
- **Relevance Filter:** Articles were screened based on title and abstract to ensure they explicitly addressed fine-grained visual tasks, multimodal alignment, or trustworthiness issues (e.g., bias, explainability) rather than general generic object recognition.

## A.4 Query Strings for Figure. 2

To quantitatively visualize the shifting research focus shown in Figure 2, we performed a trend analysis using **Google Scholar**. We queried the total number of publications per year for three distinct keywords representing the core themes of this survey. The exact query strings used were:

(1) `"fine grained classification"` — representing the traditional FGVC domain.
(2) `"Multimodal AI"` — representing the expansion into vision-language and cross-modal research.
(3) `"Trustworthy AI"` — representing the growing emphasis on reliability, fairness, and explainability.

For each keyword, we applied an exact-match search filter (using quotation marks) and restricted the results to custom date ranges for each year ($Y$) from 2013 to 2025 (i.e., as_ylo=$Y$, as_yhi=$Y$). The resulting counts were aggregated to plot the comparative growth trajectories, highlighting the stabilization of traditional FGVC research alongside the exponential surge in Multimodal and Trustworthy AI topics.

## A.5 Screening & Inclusion/Exclusion Criteria

To ensure a comprehensive and representative review of the field, we adopted a systematic literature search and screening process. The initial corpus was identified through Google Scholar using a set of carefully selected keywords related to "Fine-Grained Visual Categorization" (FGVC) and "Ultra-Fine-Grained Visual Categorization" (UFGVC), spanning multiple time ranges to capture both foundational works and recent advancements. We verified the total number of papers to ensure a robust sample size before applying our filtering criteria.

Regarding eligibility, papers were selected for inclusion if they were directly relevant to FGVC or UFGVC tasks, explicitly proposing a novel methodology, dataset benchmark, or extensive analysis, and if the full text was publicly available in English. Conversely, we excluded studies that focused solely on applications (e.g., industrial inspection) without methodological contributions, duplicate publications—retaining only the most complete peer-reviewed version—and research strictly outside the scope of visual categorization, such as non-visual modalities or pure detection tasks.

## A.6 Coding Procedure

To systematically analyze the selected literature, we developed a multi-dimensional taxonomy that characterizes how modern approaches tackle the challenges of fine-grained understanding. The coding scheme categorizes each paper across five key dimensions: the primary learning *Paradigm* (e.g., Fully Supervised, Self-Supervised, or Weakly Supervised), the *Supervision Signal* employed (e.g., Image-level vs. Part-level), the scale of *Feature Granularity*, the degree of *VLM Integration*, and the usage of *Generative Augmentation*.

*Labeling Protocol and Borderline Cases.* Every paper in our survey is assigned at least two mandatory labels: a *Task Label* (FGVC or UFGVC) and a *Paradigm Label*. To ensure consistency amidst intersecting methodologies, we established specific rules for borderline cases. For instance, papers tackling fine-grained recognition using open-vocabulary setups are coded primarily under the *Multimodal/VLM* paradigm rather than standard Zero-Shot Learning, reflecting that the core contribution typically lies in cross-modal alignment. Similarly, for hybrid architectures combining multiple supervision signals, we prioritize the signal driving the primary novelty of the proposed method.

## B Evaluation Metrics & Protocols

Evaluating fine-grained and ultra-fine-grained (UFG) visual understanding requires a shift from aggregate performance measures toward metrics that capture discriminative precision, distributional robustness, and alignment reliability. While standard classification benchmarks rely heavily on **Top-1 and Top-5 Accuracy**, these metrics often fail to reflect model behavior in real-world deployments where class distributions are highly imbalanced and semantic distinctions are subtle. Consequently, the community has increasingly adopted **Mean Per-Class Accuracy (MPCA)** as a primary metric for fine-grained tasks. unlike standard accuracy, which can be dominated by head classes in long-tailed datasets (e.g., iNaturalist[83]), MPCA weighs each category equally, exposing whether a model has truly learned to distinguish rare subpopulations or is merely overfitting to prior probabilities. For tasks involving retrieval or verification, such as identifying products in e-commerce, **mean Average Precision (mAP)** and **Recall@K** become the standard, measuring the model's ability to rank the correct fine-grained instance among visually similar distractors.

Beyond simple correctness, the requirements for Trustworthy AI necessitate metrics that assess confidence and stability. **Expected Calibration Error (ECE)**[23] is increasingly cited in FGVC literature to quantify whether a model's predicted probability scores

align with its actual accuracy, a critical property when distinguishing between ultra-fine-grained categories where visual ambiguity is inherent. Furthermore, robustness protocols have evolved to include **Worst-Group Accuracy** and **Adversarial Robustness** scores, which test model performance under specific perturbations or within the lowest-performing demographic slices. These value-added metrics ensure that high aggregate scores do not mask fragility in safety-critical or semantically specific sub-domains.

Experimental protocols in this field are categorized by how they structure the training and evaluation sets to mimic real-world scarcity. The most common setup involves **Standard Closed-Set Splits**, where training and testing classes are disjoint but drawn from the same domain, as seen in CUB-200-2011 and Stanford Cars. However, to test generalization, **Cross-Domain Protocols** are employed, where models are trained on web-scraped data and evaluated on user-captured photos, rigorously testing invariance to domain shifts. Addressing the challenge of identifying novel categories, **Open-Set and Open-Vocabulary Protocols**[17, 66] evaluate a model's ability to classify known classes correctly while rejecting or flagging "unknown" inputs. In these settings, evaluation often utilizes the Area Under the Receiver Operating Characteristic (AUROC) curve to measure the separation between known and unknown distributions. For data-scarce applications, **Few-Shot Learning Protocols** utilize episodic evaluation, typically formatted as $N$-**way** $K$-**shot** episodes [84]. Here, the model must learn to discriminate between $N$ previously unseen classes given only $K$ examples of each, testing the system's ability to acquire fine-grained concept boundaries from minimal evidence rather than large-scale statistical correlation.

The suitability of a metric is intrinsically linked to the dataset's specific challenges, particularly regarding class imbalance and label granularity. **Mean Per-Class Accuracy** is essential for long-tailed datasets like iNaturalist or fungal recognition benchmarks, where standard accuracy would allow a model to ignore the majority of rare species while still achieving high scores. Conversely, in Ultra-Fine-Grained (UFG) scenarios where inter-class differences are microscopic or subject to expert disagreement (e.g., distinct cultivars or slight disease progressions), strict Top-1 accuracy may be overly penalizing. in such cases, **Hierarchical Metrics** or **M-distance** are often more appropriate, as they penalize errors based on semantic distance in the taxonomy tree rather than treating all misclassifications as equally wrong[5]. This hierarchical evaluation is particularly relevant for datasets with open-vocab definitions, where "correctness" is better defined by semantic proximity in an embedding space than by an exact match to a discrete label ID.

## C  Limitations

Despite providing a broad overview of FG and UFGVC, this survey has several limitations. First, our literature coverage may be biased by the search strategy: the set of reviewed papers is influenced by the specific indices (e.g., digital libraries and preprint servers), keywords, and time range that we used, and we do not claim that the resulting collection is exhaustive. This also introduces *publication bias*, as preprints (e.g., arXiv) and peer-reviewed venues can differ in visibility, revision cycles, and reporting practices. Second, the proposed taxonomy necessarily involves *subjective design choices*;

boundaries between methodological categories (e.g., recognition vs. localization vs. part-based modeling, or discriminative learning vs. generative augmentation) can be ambiguous, and some works may reasonably fit multiple categories.

Third, our dataset list is non-exhaustive. The datasets summarized in Table 1 were collected through the same non-exhaustive process, so any downstream aggregation (e.g., figures derived from Table 1) should be interpreted as descriptive rather than definitive of the entire field. In particular, trends inferred from dataset counts across years may reflect discovery and selection effects, and should not be read as evidence of a sharp or universal shift in research emphasis. Fourth, performance comparisons across architectural paradigms are inherently limited: results reported in the literature are often not directly comparable due to differences in backbones, pretraining data, training recipes, data splits, annotation protocols, and evaluation metrics. Because many papers evaluate under heterogeneous settings, we avoid claiming a single global ranking of methods and instead emphasize qualitative patterns.

Finally, our discussion of evaluation practices is incomplete. While common metrics (e.g., top-1 accuracy, mean per-class accuracy, mAP, localization accuracy, calibration measures) appear throughout the literature, we do not provide a fully standardized metric taxonomy nor a unified re-evaluation across datasets and protocols. Future work could address these limitations via a fully reproducible systematic review (explicit queries, sources, inclusion/exclusion criteria, and counts), a structured cross-paradigm comparison table under controlled settings, and a more comprehensive dataset/metric audit including annotation cost, scalability, and failure modes.

