# OpenReview forum: "Fine-Grained Visual Understanding for Multimodal and Trustworthy AI"
_ACM.org/TheWebConf/2026/Workshop/TIME — TIME 2026 Poster_

### Official Review · Reviewer_HK1b · 2025-12-28
**While the paper provides a organized history and concepts, it is mostly descriptive and lacks the rigorous technical synthesis**

**Rating:** 6
**Confidence:** 4

**Review:**

The paper presents survey of FG/UFG visual understanding, reframing these tasks from classification into structural requirements for Multimodal and Trustworthy AI. The paper proposes a new taxonomy centered on the "semantic bottleneck" of visual evidence, analyzes the evolution of the field from manual features to foundation models, and provides a comprehensive catalog of datasets (table 1). Table 2 provides a useful qualitative look at datasets through "Bias/Fairness," "Long-tail," and "Explainability Signal". The main claimed contributions include perspective-driven analyses of FG/UFG, reliability amplifiers, and tools for auditing transparency in AI. However the paper seems to be shallow.
- Survey paper must analyze papers, not just list them. The paper lacks cross-comparison of performance across different architectural paradigms.
- No information was provided related to filters used to find papers to review (years, keywords, sources, count…)
- In Figure 2, the search queries used to generate these counts were not defined
- For Figure 4, it should be noted that it is derived from Table 1 (~32 datasets). As paper doesn't mention how these datasets were discovered and we don't know if it is exhaustive, Figure 4 gives narrative of a binary switch from FG to UFG in 2023
- Sections 4 and 5 consist of enumerating papers rather than synthesizing their technical contributions.
- The paper provides minimal analysis of dataset advantages/limitations (e.g., scalability, annotation cost)
- No discussion of evaluation metrics
- Missing literature review of relevant survey papers and how this paper differs from them
- No references were found to Figures 1, 2, 3
- Two dots after "Ultra-Fine-Grained Visual Categorization (UFGVC).."
- No shortcomings of this paper were mentioned
- There is no structured taxonomy/comparison table for methodologies; detailed generative augmentation investigation; zero-shot/few-shot investigation (as claimed in abstract)

---

### Official Review · Reviewer_uZix · 2025-12-31
**Fine-Grained Visual Understanding for Multimodal and Trustworthy AI**

**Rating:** 6
**Confidence:** 4

**Review:**

The authors show a historical evolution from handcrafted features to large foundation models. The paper explains that fine-grained understanding is a perceptual bottleneck where the AI misses small details.
• The survey connects subtle attribute recognition to making Trustworthy AI and Multimodal AI.
A few Points to Improve
• There are not enough public ultra-fine-grained datasets for scientists to use.
• Models sometimes use spurious correlations instead of looking at the real discriminative evidence.
• It is very hard and expensive to get expert-driven labels for these ultra-fine-grained tasks.
• Most datasets do not have ways to check for demographic bias or fairness.
• The paper should explore temporal fine-grained reasoning for videos better.
• It is difficult to integrate these tiny details into very big foundation models

---

### Official Review · Reviewer_aeuV · 2026-01-05
**Insightful survey on FGVC/UFGVC with Multimodal Models for Responsible AI**

**Rating:** 7
**Confidence:** 4

**Review:**

### Summary
This paper presents a survey focused on Fine-Grained Visual Categorization (FGVC) and Ultra-Fine-Grained Visual Categorization (UFGVC) in the context of multimodal models. It reviews relevant datasets, model evolution, and discusses future research directions.

### Strengths
- Practical Relevance: Encountering fine-grained classification tasks is very common in real-world applications. The paper addresses the critical issue of how to handle such datasets effectively.
- Structured Analysis: The paper successfully organizes and highlights critical issues in FG/UFG tasks, specifically regarding Bias, Fairness, Transparency, and Long-tail distribution problems.
- Future Directions: The authors propose several concrete research directions. These proposals are well-aligned with the workshop's goals and will likely stimulate valuable discussion among attendees.

### Weaknesses
- Lack of Criteria in Table 2: In Table 2, attributes such as Bias/Fairness are evaluated with qualitative ratings (e.g., "High", "Low"). However, the specific criteria or quantitative basis used to determine these ratings are missing. More detail on how these assessments were made is needed to interpret the table correctly.
- Redundancy: There appears to be some overlap in content between Section 6 and Section 7. The paper could be improved by consolidating these sections or clearly distinguishing their scopes to avoid redundancy.

### Questions
- In practical settings, the scarcity of expert resources and the potential for human error (even by experts) are significant challenges for FGVC/UFGVC. Does the survey or your proposed framework account for these constraints (e.g., handling noisy labels from experts or resource-efficient annotation)?

---

### Official Review · Reviewer_sNLY · 2026-01-07
**Comprehensive Survey on Fine-Grained Visual Understanding with Strong Synthesis but Non-Compliant Length**

**Rating:** 7
**Confidence:** 4

**Review:**

The paper provides a very comprehensive, integrated treatment of fine-grained and ultrafine-grained visual understanding, treating these as core requirements for multimodal trustworthy AI, as opposed to sharpening classification, which is their application within a specialist domain. The paper has a wide scope, ranging from the evolution of fine-grained and ultrafine-grained methods, a systematic survey of fine-grained data sets, multimodal alignment, to fine-grained explanations for robustness, explainability, and fairness. The structure based on perspective is very well organized, well-motivated, and has a nice focus on evaluation/deployment.

From the perspective of quality and clarity of the material presented, the work is overall well-expressed and organized. The terms related to the pivotal themes of fine-grained comprehension, ultra-fine-grained categorization, and semantic evidence are explained clearly. In contrast to the generic survey of previous work, this work attempts to integrate the trends from datasets, architectures, and the modes of evaluation, pointing to the common modes of failure with regard to the issues of shortcut learning, long-tail failure, or the lack of faithfulness in explanations. Tables of datasets and the related dimensions of trust are very informative.

In terms of originality, this is a survey paper, and its contribution lies in synthesis rather than novel algorithms. The manuscript’s originality comes from reframing FG/UFG as semantic primitives for multimodal alignment and as stress tests for trustworthiness, rather than as increasingly granular benchmarks. The integration of multimodal reasoning, trustworthy AI concerns, and deployment-oriented evaluation provides a coherent narrative that distinguishes this survey from narrower FGVC reviews.

The paper is thematically well aligned with the goals of the TIME workshop, especially its focus on evaluation beyond accuracy, interpretability, robustness under long-tailed distributions, and evidence-centric reasoning. The proposed research roadmap is thoughtful and highlights several open challenges that are directly relevant to the workshop audience.

---

### Author Rebuttal · Authors · 2026-01-13

We thank the reviewers for their constructive feedback. We have revised the manuscript accordingly, and all changes are highlighted in red in the revised submission.

---

## Reviewer sNLY

**Comment (overall):** Positive assessment of scope, organization, and framing FG/UFG as stress tests for trustworthy multimodal AI.

**Response:** We sincerely appreciate the reviewer’s supportive evaluation.

**Revision:** We made minor edits for clarity and consistency (terminology, cross-references), while preserving the overall perspective-based structure and deployment/evaluation emphasis.

---

## Reviewer aeuV

**Comment 1 (Table 2 lacks criteria):** Qualitative ratings (e.g., High/Low) need explicit criteria or quantitative basis.

**Response:** We agree. The ratings were derived from a consistent rubric based on whether each dataset/work provides explicit annotations/protocols for the corresponding trust dimension (e.g., subgroup labels + auditing for fairness; imbalance statistics + tail evaluation for long-tail).

**Revision:** We added a **clear rubric definition** (High/Medium/Low) in the **Table 2 caption and/or surrounding text**, and briefly described the evidence used to assign each rating.

**Comment 2 (Redundancy between Sec. 6 and Sec. 7):** Overlap should be reduced.

**Response:** Agreed.

**Revision:** We **re-scoped** sections: Sec. 6 now focuses on **challenges/gaps**, and Sec. 7 focuses on **actionable roadmap directions**. Overlapping text was consolidated or removed.

**Question (expert scarcity + human error/noisy labels):** Does the survey/framework account for noisy expert labels and resource-efficient annotation?

**Response:** Yes. Our framework explicitly treats expert supervision as scarce and imperfect, and emphasizes **cost-aware annotation** and **noise/ambiguity handling** (e.g., uncertainty-aware labeling, soft/partial labels, active learning/triage, and disagreement modeling).

**Revision:** We added a dedicated paragraph connecting **UFG annotation constraints** to recommended **robust learning and annotation strategies**.

---

## Reviewer uZix

**Point 1 (few public UFG datasets):**

**Response:** We agree that public UFG datasets remain limited.

**Revision:** We expanded the dataset discussion to explicitly highlight **coverage gaps**, and we added concrete directions for **dataset expansion**, including scalable protocols and community benchmarks.

**Point 2 (spurious correlations):**

**Response:** Agreed; spurious cues are a central failure mode.

**Revision:** We strengthened the discussion on **evidence-centric evaluation**, causal/robust training, and explanation faithfulness checks that reduce reliance on shortcuts.

**Point 3 (expert labeling is expensive):**

**Response:** Agreed.

**Revision:** We added a clearer treatment of **annotation cost** and strategies such as **few-shot expert validation, active selection, weak supervision, and synthetic augmentation with verification**.

**Point 4 (limited demographic bias/fairness checks):**

**Response:** Agreed.

**Revision:** We clarified fairness limitations in existing datasets and added recommended practices (subgroup metadata, bias audits, slice-based reporting, and standardized fairness evaluation).

**Point 5 (temporal fine-grained reasoning for videos):**

**Response:** Agreed that video deserves stronger coverage.

**Revision:** We expanded the section on **temporal FG/UFG reasoning**, focusing on fine-grained temporal evidence, tracking of discriminative cues over time, and evaluation protocols for temporal faithfulness.

**Point 6 (integrating tiny details into foundation models):**

**Response:** Agreed; this is a key open challenge.

**Revision:** We added discussion on **high-resolution tokenization/region detail retention**, adapter-based fine-grained alignment, retrieval/grounding mechanisms, and evaluation that probes whether foundation models truly use discriminative evidence.

---

## Reviewer HK1b

**Comment 1 (analysis vs. listing; cross-comparison across paradigms):**

**Response:** We agree that deeper synthesis is essential.

**Revision:** We added **cross-paradigm comparisons** (where reported results are comparable) and strengthened the narrative around **trade-offs** (supervision type, scalability, robustness, faithfulness). We also added a **structured taxonomy/comparison table** to summarize methodological families.

**Comment 2 (missing review protocol: filters, sources, years, keywords, counts):**

**Response:** Agreed.

**Revision:** We added a **survey methodology** subsection describing databases/sources, time window, keyword strategy, and inclusion/exclusion criteria.

**Comment 3 (Fig. 2 queries not defined):**

**Response:** Agreed.

**Revision:** We explicitly listed the **exact search queries** used to generate Fig. 2 counts.

**Comment 4 (Fig. 4 derived from Table 1; dataset discovery not exhaustive; “binary switch” narrative):**

**Response:** We agree the figure must be contextualized.

**Revision:** We clarified that the dataset collection is **representative rather than exhaustive**, explicitly stated Fig. 4 is derived from the curated dataset list, and softened language to avoid implying a strict binary shift.

**Comment 5 (Secs. 4–5 enumerate papers rather than synthesize):**

**Response:** Agreed.

**Revision:** We restructured these sections around **core themes and technical takeaways**, emphasizing shared failure modes, assumptions, and evaluation patterns rather than sequential listing.

**Comment 6 (minimal dataset advantages/limitations; scalability/annotation cost):**

**Response:** Agreed.

**Revision:** We expanded dataset analysis to cover **scalability, annotation cost, ambiguity/noise, bias metadata availability, and domain transfer**.

**Comment 7 (no discussion of evaluation metrics):**

**Response:** Agreed.

**Revision:** We added a dedicated discussion of **evaluation metrics and protocols** (e.g., long-tail metrics, open-set/OOD, hierarchical metrics, calibration/faithfulness where applicable).

**Comment 8 (missing relevant survey papers + differentiation):**

**Response:** Agreed.

**Revision:** We added a subsection comparing our survey against prior FGVC/UFGVC surveys and clarified our unique focus on **multimodal alignment + trustworthiness stress testing + deployment-oriented evaluation**.

**Comment 9 (missing references to Figs. 1–3):**

**Response:** Thank you for catching this.

**Revision:** We added explicit in-text references to Figs. 1–3.

**Comment 10 (typo: double dot after UFGVC..):**

**Response:** Correct.

**Revision:** Fixed.

**Comment 11 (no shortcomings of this paper mentioned):**

**Response:** Agreed.

**Revision:** We added a brief **limitations** paragraph (e.g., non-exhaustive coverage, comparability constraints across papers, and evolving benchmark landscape).

**Comment 12 (missing structured taxonomy/comparison; generative augmentation; zero-/few-shot as claimed):**

**Response:** Agreed.

**Revision:** We added (i) a **structured taxonomy/comparison table**, (ii) an expanded discussion on **generative augmentation** with caveats on faithfulness, and (iii) clearer coverage of **zero-/few-shot settings** (including what is currently feasible and how to evaluate it rigorously).

---

### Meta-Review · Area_Chair_9AKq · 2026-01-15

**Recommendation:** Accept (Poster)
**Confidence:** 4

**Metareview:**

This paper presents a comprehensive survey of fine-grained and ultra-fine-grained visual understanding, reframing these tasks as foundational stress tests for multimodal and trustworthy AI rather than narrow classification problems. Across reviews, there is broad agreement that the topic is timely, relevant to the workshop, and well aligned with themes such as robustness, fairness, interpretability, and deployment-oriented evaluation.

Reviewers consistently appreciated the paper’s scope, perspective-driven organization, and integration of datasets, modeling trends, and trust-related dimensions. The main criticisms focused on the initial lack of synthesis, missing methodological transparency in the survey protocol, and insufficient clarity in figures and qualitative tables. Importantly, these concerns were about presentation and analytical depth rather than correctness or relevance.

The authors’ rebuttal is thorough and responsive. They report substantial revisions, including clearer survey methodology, explicit criteria for qualitative ratings, stronger cross-paradigm synthesis, expanded discussion of evaluation metrics and limitations, and improved structure and readability. These changes directly address the most critical reviewer concerns, particularly those related to depth and clarity.

Overall, this is a solid and useful survey contribution that is well suited for discussion within the workshop context. While it does not rise to the level of a standout oral presentation due to its survey nature and limited technical novelty, it offers clear value as a reference and framing piece. I recommend acceptance as a poster.

---

### Decision · Program_Chairs · 2026-01-16

Accept (Poster)